# Graph-Based Analysis of Brain Connectivity in Multiple Sclerosis Using Functional MRI: A Systematic Review

**DOI:** 10.3390/brainsci13020246

**Published:** 2023-01-31

**Authors:** Sara Hejazi, Waldemar Karwowski, Farzad V. Farahani, Tadeusz Marek, P. A. Hancock

**Affiliations:** 1Computational Neuroergonomics Laboratory, Department of Industrial Engineering and Management Systems, University of Central Florida, Orlando, FL 32816, USA; 2Department of Biostatistics, Johns Hopkins University, Baltimore, MD 21218, USA; 3Department of Cognitive Neuroscience and Neuroergonomics, Institute of Applied Psychology, Jagiellonian University, 30-348 Kraków, Poland; 4Department of Psychology, University of Central Florida, Orlando, FL 32816, USA

**Keywords:** graph theory, network analysis, multiple sclerosis (MS), brain connectivity, functional magnetic resonance imaging (fMRI), neuroimaging

## Abstract

(1) Background: Multiple sclerosis (MS) is an immune system disease in which myelin in the nervous system is affected. This abnormal immune system mechanism causes physical disabilities and cognitive impairment. Functional magnetic resonance imaging (fMRI) is a common neuroimaging technique used in studying MS. Computational methods have recently been applied for disease detection, notably graph theory, which helps researchers understand the entire brain network and functional connectivity. (2) Methods: Relevant databases were searched to identify articles published since 2000 that applied graph theory to study functional brain connectivity in patients with MS based on fMRI. (3) Results: A total of 24 articles were included in the review. In recent years, the application of graph theory in the MS field received increased attention from computational scientists. The graph–theoretical approach was frequently combined with fMRI in studies of functional brain connectivity in MS. Lower EDSSs of MS stage were the criteria for most of the studies (4) Conclusions: This review provides insights into the role of graph theory as a computational method for studying functional brain connectivity in MS. Graph theory is useful in the detection and prediction of MS and can play a significant role in identifying cognitive impairment associated with MS.

## 1. Introduction

Multiple sclerosis (MS) is an autoimmune inflammatory disorder that occurs in the central nervous system (CNS) and, in contrast to other neurological diseases, primarily affects young people. MS refers to the demyelination of myelin sheaths due to the immune system’s attack of the CNS [1,2]. Disabilities resulting from MS in young people have deleterious lifestyle effects and often restrict physical activity. Other MS symptoms include difficulty in walking and controlling body movements, fatigue, vision problems (blurry vision), and low cognitive performance [3]. People with MS are more likely than unaffected individuals to experience depression [4,5,6]. Moreover, some patients in advanced disease stages (high Expanded Disability Status Scale (EDSS) scores) experience impaired cognitive function, such as difficulty in recalling memories. In some cases, vocal fluency is affected [7]. The EDSS categorizes patients according to their disabilities on a scale of 0 (no/minor disability) to 10 (death due to MS) [8].

Although MS is widely believed to be a disease affecting western or northern countries, recent studies have revealed that MS is, in fact, a global disease, and substantial evidence has indicated a high incidence in the eastern Mediterranean [9,10,11]. Some findings have suggested that the incidence of MS may be due to a lack of vitamin D. Therefore, MS risk factors include sunlight exposure, genetic factors, and other environmental factors [12]. Recent findings have revealed that Epstein–Barr virus (EBV) is a major factor that may increase the risk of MS, by as much as 32-fold [13].

There are several clinical MS phenotypes, as follows. Clinically isolated syndrome (CIS) defines the initial episode of MS but may not meet all criteria for MS. Many people with CIS have CNS demyelination and may show disease progression to more advanced stages. Relapse/remitting MS (RRMS) involves deterioration of neural function and includes occasional relapses (85% of all patients). Patients with RRMS show no new signs after experiencing a period of relapses. Secondary progressive MS (SPMS) may develop in patients who progress from RRMS with more severe symptoms. Progressive relapsing MS (PRMS) is characterized by neurological malfunction without early relapse or remission [1,14].

EDSS scores are used to identify the extent of MS progression. This scale categorizes patients on a scale ranging from 1, indicating minor signs of disability/impairment in a single function, to 10, indicating death due to MS [8].

The McDonald criteria are widely used to diagnosis MS [15]. The number of attacks and the number of clinical lesions is the basis for MS diagnosis [16]. For an early-stage diagnosis, cerebrospinal fluid is among the most critical factors. In contrast, diagnosis for disease progression involving brain lesions, spinal cord lesions, and positive visual evoked potential should be considered [15]. EDSS (Expanded Disability Status Scales) is used to identify the level of MS progression. This scale categorized patients from 1 with small sign of disability/impairment to 10 (death to MS) [8].

Neuroimaging techniques are used to study neurological disorders including Alzheimer’s disease, epilepsy, and Parkinson’s disease [17]. As previously described, some symptoms can be used to diagnose MS. However, many symptoms, such as weakness, fatigue, and difficulty in movement, are common to multiple neurological disorders. One method to increase diagnostic accuracy is precise neuroimaging, which provides compelling evidence of any malfunctions in the nervous system [18]. Several neuroimaging techniques are applicable in MS studies. fMRI is a neuroimaging technique that can be used when high spatial resolution is required to make an accurate diagnosis. Lesions are located in the periventricular, juxtacortical, and infratentorial regions. Because of its high spatial resolution, MRI/fMRI is an appropriate technique for studying the effects of lesions on brain connectivity and can enable faster diagnosis. fMRI has been applied to assess movement- and cognitive-associated impairment in MS patients [19].

The role of quantitative and computational approaches is indisputable in any field of research involving diagnoses of conditions such as cancer and nervous system diseases [20,21]. In diagnosing these types of diseases, including MS, the timing of diagnosis is important. The use of computational approaches may lead to rapid diagnoses. Timely diagnoses result in more effective treatment and medication applications to prevent the progression of the disease. This review presents graph theory applications in MS studies. Studying brain functional connectivity is critical to identify biomarkers for MS diagnosis. In recent years, computational methods, including machine learning and deep learning, have been widely applied to MS data [22]. In applying them, there are some limitations that may affect the result including, but not limited to, the high impact of artifacts and MRI image quality in the results [23]. In brain functional connectivity studies, graph theory is one of the best methods for determining the activation of regions and interactions between regions [24]. Graph theory considers interactions between nodes (vertices), and the analysis of correlations between regions provides insights into regional activation and alterations. Another advantage of applying graph theory in the MS field lies in the idea of “economy of brain network organization”. According to this, MS is a neurological disorder in which the longest connections in the network have been influenced by the disease [25]. Graph theory, by studying interactions between regions, may represent network alteration for patients with MS.

This paper aims to review the application of graph theory in MS studies based on fMRI. To accomplish these objectives, we reviewed the applications of neuroimaging and computational modeling in MS. We focused on applying graph theory as a computational method and fMRI as a neuroimaging technique. We applied PRISMA guidelines to identify relevant articles systematically. Then, we discussed applications of graph theory in MS diagnosis to the disease progression tracking. This systematic review was performed to highlight the important role of graph theory in MS studies.

## 2. Materials and Methods

The present systematic review applied the Preferred Reporting Items for Systematic Reviews and Meta-Analyses (PRISMA) approach [26], as shown in Figure 1. The PRISMA guidelines were formulated to guide the collection of related articles to create a reliable systematic review. To accomplish this, duplicate articles from different databases are excluded as instructed in the PRISMA.

### 2.1. Research Questions for Literature Review

After reviewing previous studies in this field, we aimed to address the following research questions (RQ) in the literature review:RQ1:What is the role of fMRI in studying brain connectivity in people with MS?RQ2:How is brain connectivity altered in the brain network in people with MS?RQ3:What is the value of applying graph theory for neurological disorders such as MS?RQ4:How does graph theory help detect early stages of MS or MS progression?

### 2.2. Study Search and Selection Process

In the first step, various databases were used to find relevant articles. Six categories of keywords were searched, and the PRISMA guidelines were followed in finding relevant articles. The population, modeling, outcomes, and conditions were considered to avoid bias. Different resources were used to search the content, including journal articles, textbooks, proceedings, and published conference proceedings. Many search engines were used in this study, including Google Scholar, IEEE Xplore, PubMed, EBSCO HOST, Web of Science, Scopus, and Science Direct. The six categories of keywords were used to find relevant studies as follows:“Multiple sclerosis” OR “MS” AND “fMRI” AND “graph theory”;“Multiple sclerosis” OR “MS” AND “fMRI”“Multiple sclerosis” OR “MS” AND “graph theory”;“Multiple sclerosis” OR “MS” AND “brain connectivity”;“Multiple sclerosis” OR “MS” AND “brain network”;“Multiple sclerosis” OR “MS” AND “graph theory” AND “node”.

### 2.3. Eligible Studies and Quality Assessment

The following eligibility criteria were used to select articles for this systematic review:Articles in English, original peer-reviewed articles,articles collecting functional magnetic resonance imaging (fMRI) data,articles with precise methods and explanations of the results,articles using graph theory for the analysis.

The following exclusion criterion was used:Articles not answering or aligning with the research questions.

Quality assessment was performed by both reviewers (S.H. and W.K.), and the Risk of Bias in Systematic Reviews (ROBIS) approach was applied [27]. This approach has three phases: considering the relevance of the studies, finalizing the review process, and assessing the risk of bias.

## 3. Results

Taking keywords into account, we identified 2445 journal and conference articles in the first step of the PRISMA guidelines. Because most relevant articles were found via Google Scholar, few duplicate articles were detected in the other sources (n = 450). Another reason for the few duplicate articles was that the other database results for the keywords used in the present literature review were limited. For instance, the results for one set of keywords (“multiple sclerosis” or “MS” and “graph theory” & “fMRI”) in the Web of Science yielded only 32 articles. After reviewing the abstracts of the identified articles, only 152 articles were selected according to the inclusion criteria, such as articles in English, articles with fMRI applied in the methods, and articles using graph theory in the analysis. In the screening stage, full-text articles were searched to identify eligible articles. As shown in Figure 1, a total of 24 articles aligning with our research questions and with all inclusion criteria were identified. 

### 3.1. Quality Assessment of the Selected Articles

As described in the methods section, ROBIS was followed by both reviewers (S.S. and W.K.) in the present literature review [27]. According to the ROBIS guidelines, the current literature review was categorized under the diagnosis and prognosis review type. In reviewing article types, patients, outcomes, the intended use of the model, and conditions should be identified. In Table 1, the sample size for each article, findings, EDSS, experiment types (conditions for experiments), and graph properties for each specific study are shown. As shown in Figure 2, among all articles included in this systematic review, 20 were of good quality, met the inclusion criteria, and had low bias; the remaining articles (16% of all articles included in this study) had a high risk of bias.

### 3.2. Study Characteristics

As shown in Table 1, the number of participants ranged from 8 to 332 (mean = 75, median = 37.5). Among the selected articles in this field, 87% used a healthy control group to study the effect of the disease on brain network connectivity. All selected articles focused on patients with EDSS scores below 7.5 because patients with higher EDSS, owing to movement limitations, have difficulty participating in fMRI studies. The database searching was conducted from 2000 to the present. However, all articles included in this systematic review were published between 2009 and 2021. The publication trend for the articles included in this review is shown in Figure 3B. The increase in the use of graph theory is illustrated by the number of publications from recent years.

### 3.3. Overview of the Review

To answer research questions 1–4, the value of fMRI in MS studies must first be understood; therefore, we provide a review of neuroimaging techniques in this field (RQ1). We then give a brief review of brain connectivity and the role of graph theory in studying functional and effective brain connectivity for patients with MS (RQ2). To understand the applications of graph theory in MS studies, we provide an introduction to graph properties (RQ3). In the results and discussion, we (S.H. and W.K.) discuss how graph theory can help detect the incidence of MS or disease progression (RQ4).

## 4. Discussion

Since the main focus of this paper is to review the application of graph theory to fMRI data acquired from MS patients, the following topics are discussed in this section:The role of fMRI as a neuroimaging method in studying MS;Brain connectivity in MS (including brain functional connectivity and brain effective connectivity in MS);The role of graph theory as a computational approach in studying MS (including referring to other computational approaches, graph theory for disease detection, disease progression, and network alterations in MS);Limitations and considerations of applying graph theory to fMRI data on MS patients.

### 4.1. fMRI in MS Studies

As shown in Figure 3, several neuroimaging techniques, such as fMRI, fNIRS, EEG, MEG, and DTI, were applied to investigate different impairments in MS, including attention, walking, and diagnosis (Figure 3A). fMRI has been used in various studies of neurological disorders, including multiple sclerosis, epilepsy, and Alzheimer’s disease. In the field of MS, owing to its high spatial resolution, fMRI has been applied more than other neuroimaging techniques when graph theory is chosen to model the network (Figure 3B). Two types of fMRI are used in MS studies: resting-state fMRI (applied to study a specific part of the brain in resting state) and task-based (including a particular task pertaining to topics such as plasticity and memory) [22,53]. In task-based studies in patients with MS, a designed task, such as a paced auditory serial addition test (PASAT), is performed during fMRI recording [7,54,55]. The increase in published articles in the MS field applying graph theory to fMRI data is shown in Figure 3C.

**Figure 3 brainsci-13-00246-f003:**
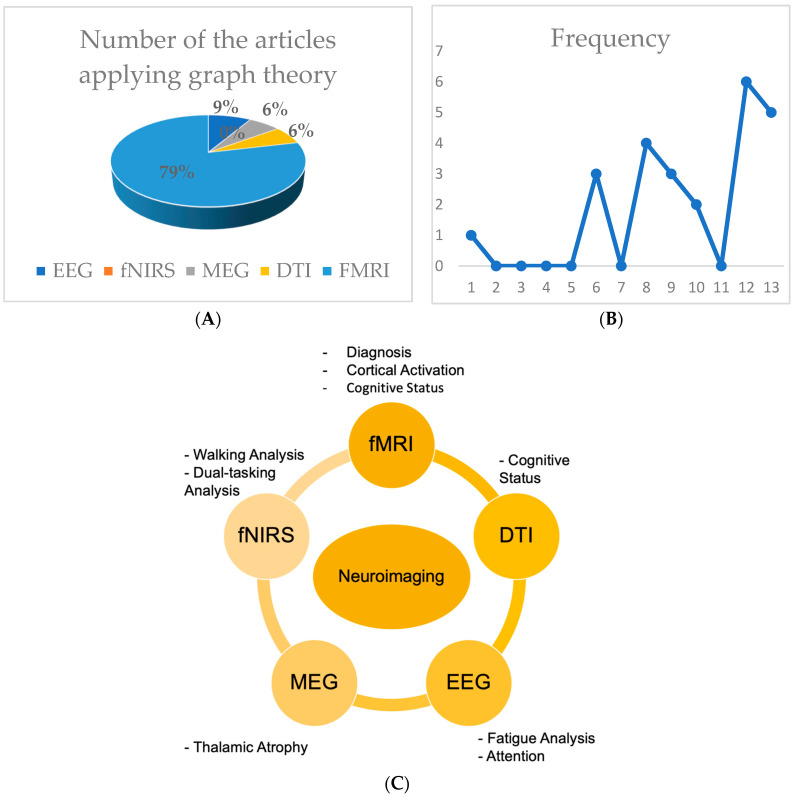
Graph Theory and neuroimaging: Several neuroimaging methods can be applied for studying MS, such as functional near-infrared spectroscopy (fNIRS), EEG, MEG, DTI, and fMRI. Patients with MS experience difficulties in walking. fNIRS is applied primarily in walking analysis studies and multi-tasking studies, including physical and mental activity [56,57,58]. In addition, fNIRS, because of its capability of measuring hemodynamic response in dual tasks, is used in working memory tasks in patients with MS [59]. Electroencephalography (EEG) is used for motor-cognitive impairments to aid in the study of gait kinematics with high resolution [60]. Early physical fatigue, a sign of MS, is studied primarily with EEG and fNIRS [61,62]. According to EEG data, higher beta temporal-parietal functional connectivity (in the resting state) is a possible sign of early fatigue in MS [63,64]. Diffusion tensor imaging (DTI) is an important neuroimaging technique to investigate cognitive status in patients with MS at the macro and micro levels [65]. Moreover, magnetoencephalography (MEG) is another neuroimaging technique applied to MS data. In some studies, MEG has been performed along with fMRI Combining these methods provides better insight into the extent of integration in the brain functional network [66,67,68]. MEG has been applied to investigate thalamic atrophy [68,69]. (**A**) Classifying neuroimaging techniques applying graph theory in MS; (**B**) Frequency of papers that apply graph theory and fMRI in MS field included in this review; (**C**) Application of different types of neuroimaging in MS studies.

### 4.2. Brain Connectivity in MS

Functional and structural connectivity were applied to MS data [70]. Nodes and edges illustrate inter-neuronal and regional connectivity [71]. As shown in Table 2, alterations in functional connectivity density under disability progression have been observed [33]. Structural–functional coupling is an approach to visualize the progression of MS that can serve as an index in diagnosing early-stage MS [48].

In addition to functional connectivity, which we discuss below, specific thalamic and hippocampal connectivity was examined in MS studies. Thalamic connectivity studies have shown that damage to the grey matter nuclei can be traced to track cognitive function in patients with MS [7]. Thalamic connectivity increases in the resting state when cognitive performance is limited by severe MS [72,73]. The more severe the disease, the greater the thalamic functional connectivity [73]. Frontoparietal and occipital brain regions have been reported to show changes in parallel to thalamic connectivity in patients with MS [72].

**Table 2 brainsci-13-00246-t002:** Regions with altered functional connectivity in MS.

Papers	Regions with Altered Functional Connectivity in MS	Alteration in FC
[74]	Bilateral inferior parietal cortex, posterior cingulate, medial prefrontal cortex	
[75]	Caudate nucleus and motor cortex	
[76]	Anterior and middle parts of the putamen, adjacent globus pallidus, anterior and posterior thalamus, subthalamic nucleus, motor network	
[77]	Sensorimotor and somatosensory association	
[78,79]	Right sensory-motor and premotor cortex, and anterior cingulate gyrus	
[80]	Subcortical and default-mode networks	
[81]	Insula and the dACC to the striatum	
[82]	Thalamus and several cortical regions	
[82]	Intra- and inter-thalamic connectivity	
[83]	Left hippocampus and cortical regions	
[84]	Medial prefrontal and frontal pole regions	
[85]	Fusiform gyrus with the right lateral occipital cortex	
[86]	Sensorimotor cerebellum with cerebellar, thalamic, and cortical (frontal, parietal, and temporal) areas	
[86]	Sensorimotor cerebellum with insular areas	
[87]	Right medial thalamic nuclei connectivity with bilateral caudate/thalamus and left cerebellar areas	
[87]	Left anterior thalamic nuclei and anterior cingulate cortex, bilaterally	
[88]	Superior ventral striatum and posterior cingulate cortex	
[89]	DMN in the posterior cingulate cortex and bilateral inferior parietal cortex	
[72]	Bilateral hippocampal and dorsal-frontal components	
[72]	Cerebellum, thalamus, cingulum, and prefrontal cortex components	
[33]	Prefrontal cortices, anterior and posterior cingulate cortices, anterior and middle temporal cortices, inferior parietal cortex, and thalamus	

The hippocampus has a key role in memory. Previous studies have suggested that hippocampal connectivity is also associated with disease severity and loss of hippocampal volume, resulting in verbal memory impairment [48,90,91]. As hippocampal activation decreases, connectivity increases in the brain network [92].

### 4.3. Brain Functional Connectivity in MS

In MS studies, model-free and model-based functional connectivity were broadly applied. The correlation coefficient was used as a model-based method in several MS studies (Figure 4) [93]. Correlations between the brain network and cognitive impairment illustrate the disease course or progression. As depicted in Figure 5, The severity of cognitive impairment caused by MS is reflected by a significant increase in functional connectivity in the default-node network [74]. Therefore, functional connectivity in MS reflects the disease course. Other findings revealed that, although an increase in functional connectivity was expected in the early stages of MS, functional connectivity enhancement decreases with disease progression [33]. In studies of fatigue in MS, functional connectivity was found to increase in the caudate nucleus and the motor cortex under fatigue conditions [75]. Another study revealed sex-specific alterations in functional connectivity in MS [94].

### 4.4. Brain Effective Connectivity in MS Studies

Effective connectivity provides insight into how a given region in the brain affects other brain regions (Figure 4). In MS studies, any abnormalities in the connections between regions can be inferred as cognitive and physical impairments [97]. In studies on patients with MS, effective connectivity showed that the frontal cortices have high connectivity with multiple regions [99]. Few studies in the MS field applied effective connectivity to study brain networks. The limited number of articles using EC may result from the limitations of EC in previous articles, such as the existence of a priori assumptions regarding the interaction between two regions [97].

### 4.5. Graph Theory in Studying Brain Connectivity

Graph theory is a mathematical approach to studying network connectivity. In neuroscience, it is widely applied to study interactions between regions. In this model, nodes represent regions, and edges represent the interactions between regions (Figure 6).
(1)$$\mathrm{Eglob}\;(G)=\frac{1}{N\;\left(N-1\right)}\sum_{i\neq j\epsilon G}\frac{1}{\mathrm{Lij}}$$
(2)$$\mathrm{Eloc}\left(G\right)=\frac{1}{N}\sum_{i\epsilon G}\mathrm{Eglob}\;\left(\mathrm{Gi}\right)$$

### 4.6. Graph Theory in MS

In this study, our focus was on the application of graph theory to fMRI data from patients with MS (Figure 7). Blood oxygen level-dependent fluctuations in resting-state fMRI provide insights into brain connectivity in the entire brain [33]. Graph theory analysis identifies patterns in brain connections and investigates structural and functional systems [71]. Any dysconnectivity may reveal the brain network for particular neurological disorders such as MS. Abnormalities can be demonstrated through graph theory in patients with MS [71]. Several parameters have been used to study early MS impairments with graph theory parameters, such as centrality, segregation, and integration [48]. Functional and structural connectivity was applied to study changes in whole-brain networks [93]. Moreover, serotonergic, noradrenergic, cholinergic, and dopaminergic systems may be examined to investigate fluctuations in functional connectivity [41]. One benefit of functional connectivity is helping researchers define specific biomarkers to classify disease impairments and severity [39]. In contrast, the graph–theoretical approach is used to detect cognitive dysfunction by classifying task-based brain functions [102]. Node-based graph theory also helps researchers detect specific regions and functions in patients and healthy controls [37,64]. Brain connectivity, along with the nodal degree between and within different regions, was studied to determine the sources of impairment in patients with MS [34]. Functional connectivity in working memory studies using fMRI has resulted in identifying a biomarker for detecting early-stage MS [29].

#### Properties for Graph Theory Using MS Data Based on fMRI

Global and local networks were applied to investigate the effects of brain connectivity or impairments. The results of previous studies suggested a lower global efficiency in patients with MS than healthy control individuals [32,47]. One graph property is small-worldness, which refers to a condition in which nodes are neighbors to each other but are still connected through other nodes with a number of edges and nodes [103]. Small-world efficiency is used for structural and topological characterization [65]. This property was suggested to provide an economic brain functional network [104]. Figure 7 shows the overall concepts and applications of node-based analysis. Figure 6 depicts the application of each graph theory property in MS studies. Global efficiency, local efficiency, network modularity, clustering coefficients, and centrality are frequently used in the MS field.

### 4.7. Other Computational Approaches in MS Studies Using fMRI

With the advent of computer-aided diagnosis (CAD), several computational methods have been applied in fields as diverse as disease detection and patient classification. Among all computational approaches, graph theory, machine learning, deep learning, and Bayesian modeling have been extensively applied to MS data. Machine learning is among the best tools applied to diagnose the early stages of MS [22,105]. In this respect, several algorithms have been used, such as Random Forest, Support Vector Machine (SVM), General Linear Model, partial least squares, neural network, naïve-Bayes, and K-nearest-neighbor [22,106,107,108]. These algorithms have been applied in classifying biomarkers and predicting the disease course. Deep learning approaches in MS have been applied to classification, segmentation, and detection [109,110,111,112]. According to the McDonald criteria, deep learning is also used to find the exact locations of brain lesions. DL is a CAD approach for patients with MS [23]. The Bayesian model has been used as a computational model incorporating the basics of probability studies. Studying risk factors, including sex and the environment, are examples of applying the Bayesian model in MS studies [113]. We did not find any specific articles after adding “fMRI” to the keywords in our search for the Bayesian model, we did not find any specific articles. The nature of the Bayesian model helps researchers study underlying causes and probabilities of the disease course or disease progression. Therefore, MRI studies usually do not apply the Bayesian model to study brain networks.

#### Graph Theory Combined with Machine Learning in MS Studies

Mathematical and statistical approaches are inseparable methods used in analyzing brain connectivity. General linear model is a mathematical method applied in graph-theoretical studies in MS to detect brain activation [102,114]. Some articles in recent years have applied a combination of graph theory and machine learning. Various machine learning classifiers such as SVM have been used to distinguish patients with graph metrics from healthy individuals. SVM is a supervised machine learning approach for classifications that aims to find the best algorithms for classifying classes or groups of specific data.

Moreover, the use of SVM on graph-based measures has enabled the investigation of biomarkers to distinguish people in early disease stages from healthy controls [49]. The use of SVM can avoid or decrease the effects of overfitting of the results and consequently the introduction of bias [115,116]. Previous studies have shown that SVM increases model accuracy and can be applied to MS data-driven graph parameters. Moreover, the combination of graph theory and SVM enables the identification of more sensitive biomarkers that can distinguish different types of MS, such as RRMS and PPMS [101]. Because the graph properties are sensitive, and SVM classification is accurate, this method may provide a valuable tool for disease detection in patients with MS.

### 4.8. Disease Detection and Prediction in MS Studies with Graph Theory

No cure exists for MS, although the disease can be controlled. Therefore, identifying the disease in the earliest possible stages is essential. Most articles in this review included healthy control individuals in their experiments to enable fMRI image comparison. However, network properties could also be used as an indicator in disease detection and the prediction of disease progression, as illustrated by the modularity values in the network for patients with MS. Higher modularity can be inferred as an indicator of MS in the brain network [29,34]. In parallel, a decrease in centrality is seen in the ventral stream and sensorimotor regions in the entire brain network in patients with MS [31]. DMN centrality was suggested to be greater in patients with MS [36]. Local connectivity is more significant in patients with MS than in the HC network. However, the strength of the connections is lower in the early stages of MS [34,39]. Another approach to identify the early stages of the disease involve cognitive tasks. Dual tasking experiments and fatigue in patients with MS cause alterations in the network [28,30]. Different network hub formations in the cerebellum and left temporal region were reported in the networks of patients with MS in early stages [32].

#### 4.8.1. Network Abnormalities in Patients with MS

Abnormalities in the global and local networks are found in patients with MS [32]. A significant abnormality is a reduction in default-mode, frontoparietal, and visual networks dynamics in patients with MS [117]. Moreover, the neuro-modularity network shows abnormalities in functional connectivity in patients with MS, and centrality is decreased in the brainstem for the serotonergic–noradrenergic network but increased in the cerebellum for the same network [41].

#### 4.8.2. Cognitive Impairment and Network Efficiency in MS

The focus of this section is to explore the effects of cognitive impairment in patients with MS on network efficiency. Studies have suggested that cognitive impairment contributes to a decrease in the nodal degree (i.e., the number of edges neighboring a given node) in the bilateral caudate nucleus and right cerebellum in MS [101]. The increase in thalamic connectivity, in contrast, is correlated with cognitive impairment [72]. Alterations in DMN are critical to evaluate the efficiency of the network. The results of previous studies shed light on these reduction dynamics in DMN and indicated that the frontoparietal and visual networks are associated with cognitive impairments in MS [36,95,118]. The robust evidence indicating the role of cognitive impairment in this alteration is the increase in connectivity in DMN in patients with CIS who are not cognitively impaired [119]. Notably, decreases in network efficiency and small-worldness were identified in patients with MS with cognitive dysfunction [42].

### 4.9. Graph Theory Analysis of Disease Progression in MS

Studying brain networks in MS provides insight into cognitive and physical functions that may optimize disease progression evaluation. Recent studies revealed that disease progression may decrease the connectivity in the brain network in patients with MS [33].

Hubs (High-Degree Nodes)

According to previous studies, hubs are assumed to reveal important information regarding the effects of neurological disorders on brain functional connectivity. This finding was suggested to be rooted in a decrease in inhibitory activity in the brain network, particularly in hubs [64,96]. Although hubs are not a reliable source for studying the very early stages of MS, hubs as core regions are sensitive to MS progression [96]. Functional connections are negatively affected during disease progression. In contrast, lost hubs have been reported in the thalamus, left frontal lobes, superior frontal gyrus, precuneus, and anterior cingulum in the left hemisphere in patients with MS [7,32]. In addition, cognitive impairment was revealed in hub analysis of MS progression [7]. The rich club is another term that has been used in MS studies using graph theory. It is defined by high degree/strongest nodes [37]. A decrease in rich club connections was observed in patients with PPMS [37]. In patients with MS, higher physical disability is associated with lower connections in rich-clubs, and cognitive impairment is associated with loss of rich hubs [37,64].

White Matter Lesions

Changes in white matter lesions in patients with MS affect high-degree regional connections. Some studies divided patients into subgroups based on white matter lesion load [28]. A disruption in small-worldness is associated with white matter lesions [28]. White matter and grey matter changes are used as biomarkers in the detection process and in identifying disease progression. White matter lesion accumulation has a substantial role in disease progression, which could be addressed in changes in functional connectivity in patients with MS [33].

### 4.10. Toolboxes for Statistical Analysis in MS Studies Using fMRI

The Brain Connectivity Toolbox (brain-connectivity-toolbox.net) was implemented in some studies to measure network properties and detect modularity maximization [29,32,39,42,45,46,48,49,50]. The CONN toolbox is commonly used in MS studies applying graph theory to study functional connectivity [34,41]. Brainwaver is another toolbox that allows researchers to measure brain connectivity [33]. GAT and GRETNA are utilized to analyze brain connectomes and measure network properties [34,44]. DPARS (Data Processing Assistant for Resting-State fMRI) is an SPM-based toolbox for investigating functional connectivity and network measures in MS studies [40,120].

### 4.11. Limitation and Future Directions in MS Studies Using Graph Theory

Experiment Design:

PASAT is used in working memory and attention studies of MS, and the Stroop task is designed to trace the sensorimotor cortex [2,7,29,38,55,72]. Although many studies applying fMRI have utilized PASAT, this experiment is more applicable to working memory studies and cannot directly indicate neuropsychological function. Applying multiple experiments in future studies may lead to a comprehensive understanding of all neuropsychological functionalities for MS patients.

Sample Size

Small sample sizes pose another limitation in MS studies, given the difficulties involved in conducting experiments with all considerations for patients with MS [10,22,44,115,121]. Small sample sizes may affect the results of studies, and increasing the number of participants will improve analyses in graph theory. This is a potential direction for future studies in MS. 

Graph-Theory-Related Limitations in MS Studies

In graph theory, Pearson’s correlation is a linear bivariate model. Therefore, this model cannot fully reveal the complex connectivity in the MS brain network [10,47]. In addition, in prior studies, researchers selected the voxel size and parcellation methods without any optimum modeling reflecting the thickness and convolutions [10,40,46,115]. In future work, the voxel size could be optimized to improve the analytic results.

Network Reliability Limitations

There are many factors that may affect the reliability of the network, including, but not limited to, test–retest data set, different atlases, thresholding, and parcellations. As mentioned, one side of the reliability goes back to the data collection of different sessions that may lead to different network measures [42]. In addition to that, Different atlases, parcellation approaches, and thresholding should be considered in measuring the reliability of the network [122].

Study Design Limitations

Experimental designs in MS studies have several limitations. Functional metrics in cross-sectional design should be considered when studying patients with MS with different disease courses [10,33,50]. As another limitation of the cross-sectional design, some features cannot correctly demonstrate functional abnormalities [32]. Temporal dynamics and related cognitive impairments are missing in cross-sectional designs. To tackle this issue, cognitive test monitoring would be useful in future studies. Moreover, the reverse inference approach was previously proposed to address these missing components for interpretations and analyses of cognitive process [123]. As suggested in this approach, evidence is needed to link changes in interactions of specific regions to cognitive impairment. This evidence might be provided by cognitive tests.

Disease Classification

A lack of study participation among patients in very severe stages of disease presents another limitation in MS studies [44]. Most participants have a low or moderate disability, resulting in a gap in the network [107]. Moreover, the data cannot be validated because of a lack of longitudinal data [40,96]. In future studies, including MS patients from various EDSSs, ranging from low to high, could give researchers greater insight into the physical and cognitive impairments caused by MS. Such information would help in identifying accurate biomarkers for diagnosing MS progression.

Overlapping in the Network

In contrast to the neuro-modulatory network in MS studies, overlapping in the non-modulatory network may show functional connectivity alterations among other regions. However, such overlaps do not necessarily depict alternations in functional connectivity [10,41]. Region of interest (ROI) selection might be a good approach for reducing overlaps in brain regions in the neuro-modulatory network [41]. Thus, overlapping in the modulatory network is a limitation in MS studies that could be addressed in future works.

Suggestions and Future Work

MS is a neurological disorder that negatively affects the longest (or more expensive) connections or edges in the entire system based on the economic connectivity framework [104]. However, in the MS field, edges have not been used for analysis, although they have been considered in other fields such as autism [25,96,124]. Studying edges plus nodes as the main elements of the network could provide more information on functional connectivity in the brain networks affected by MS.

Another appealing idea for a future paper in the MS field would be meta-analysis approaches for graph theory metrics on MS data. Previously, it was applied for brain injury data, depression, and other neurological disorders [125,126].

Recently, the multi-model network was extensively used to study neurological disorders. This approach may give researchers a comprehensive view of the whole network when neuroimaging data and analysis is combined with cognitive tests and data or questionaries collected from patients [42,127]. More multi-model studies are needed in the field of MS to grasp a better insight into network alterations.

In recent years, a notable approach to classifying data from fMRI is graph neural networks. This approach was first developed to graph convolutional neural networks between 2009 and 2016 [128,129]. It has since been used in many neurological disorders; however, few studies have used graph neural networks in the MS field [130,131].

## 5. Conclusions

In this systematic review, we discussed the role of graph theory in detecting MS and disease progression, including the effects of cognitive impairment on network efficiency in MS. This knowledge should help researchers model disease progression in patients with MS because brain network connectivity may reflect the brain’s cognitive performance. Global and local efficiency are frequently applied to evaluate brain networks in patients with MS. However, graph theory has several limitations that stem from the sample size, experimental design, and the challenges of overlap in the brain network. Our review of previous studies highlighted the importance of alterations in networks such as the thalamus and DMN. Increasing the sample size and applying combined computational approaches might provide more accurate network modeling results in future studies for patients with MS. To conclude, our review of applying graph theory in MS studies revealed that graph theory as a computational method helps researchers better understand brain connectivity changes in patients with multiple sclerosis.

## Figures and Tables

**Figure 1 brainsci-13-00246-f001:**
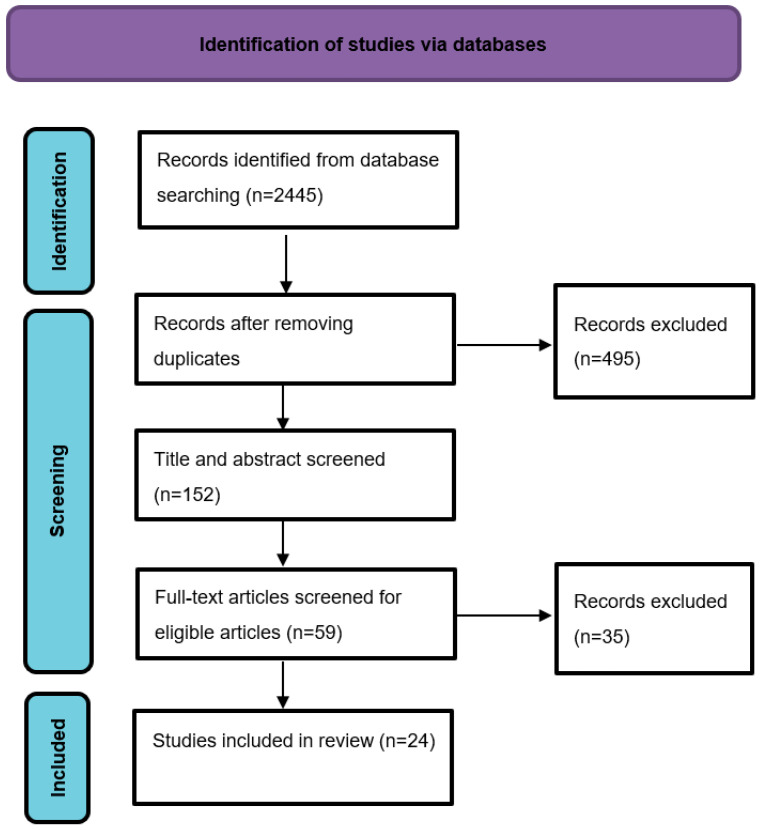
PRISMA diagram for the present literature review.

**Figure 2 brainsci-13-00246-f002:**
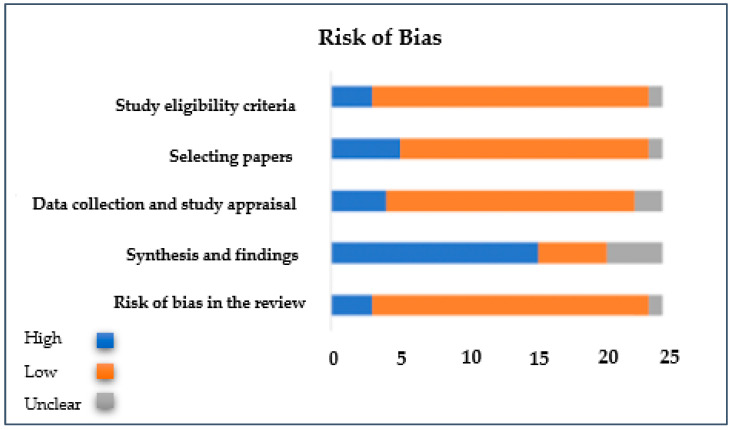
ROBIS approach for identifying bias in the present review.

**Figure 4 brainsci-13-00246-f004:**
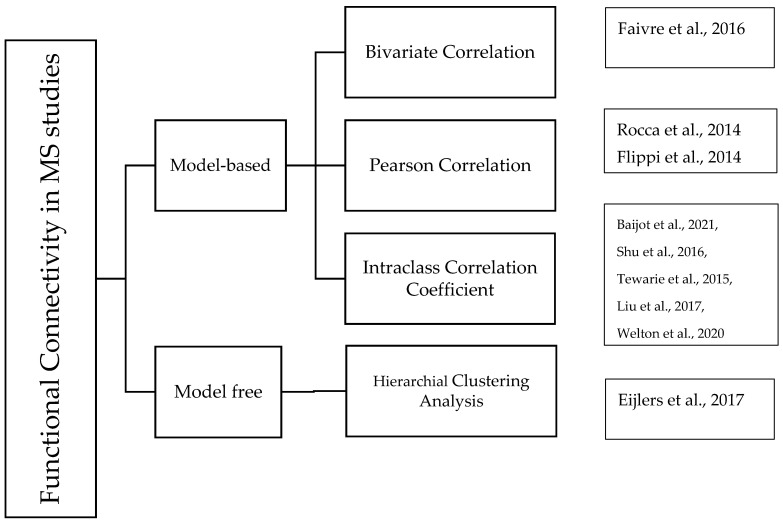
Literature on functional and effective connectivity in MS studies. [2,30,33,36,38,42,47,66,95,96,97,98,99,100].

**Figure 5 brainsci-13-00246-f005:**
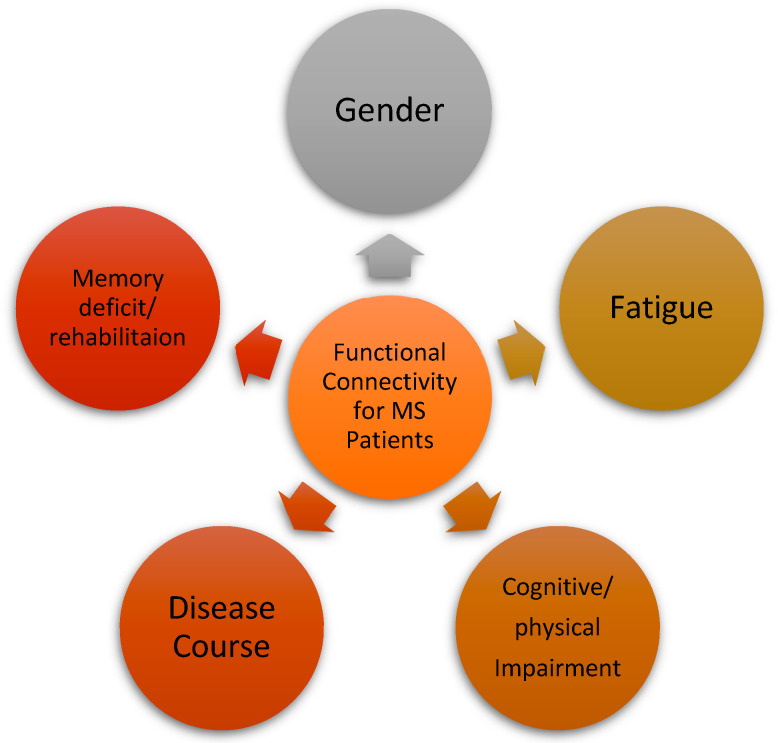
Clinical factors impacting the functional connectivity.

**Figure 6 brainsci-13-00246-f006:**
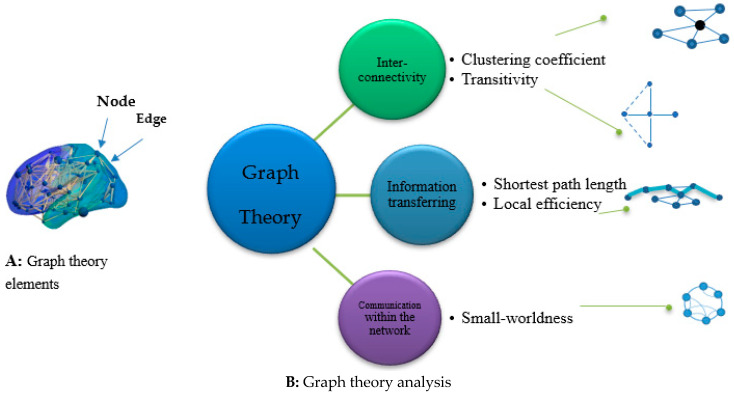
(**A**) Graph theory elements. (**B**) Graph theory analysis Clustering coefficients: These coefficients measure segregation in the network and the capability of interconnections for a specific node. Transitivity: Transitivity is the proportion of the number of the triangles in a specific region and the total number of triangles [101]. Global efficiency: Global efficiency is a parameter to measure segregation and integration of the brain network [96]. Equation (2) is used to calculate global efficiency in a brain network, where Lij is the shortest path between node i and node j [65]. (Equation (1), global efficiency). Local efficiency: Local efficiency calculates short distance connectivity [33]. (Equation (2), global efficiency). Small-world network: This network is named according to its origins in the social sciences. The randomness of a network is assessed with the small-world parameter, which reflects the effectiveness of the transformation of information within the network. Studies in MS have shown that small-world efficiency is significantly diminished in patients with MS compared with control individuals [28,65].

**Figure 7 brainsci-13-00246-f007:**
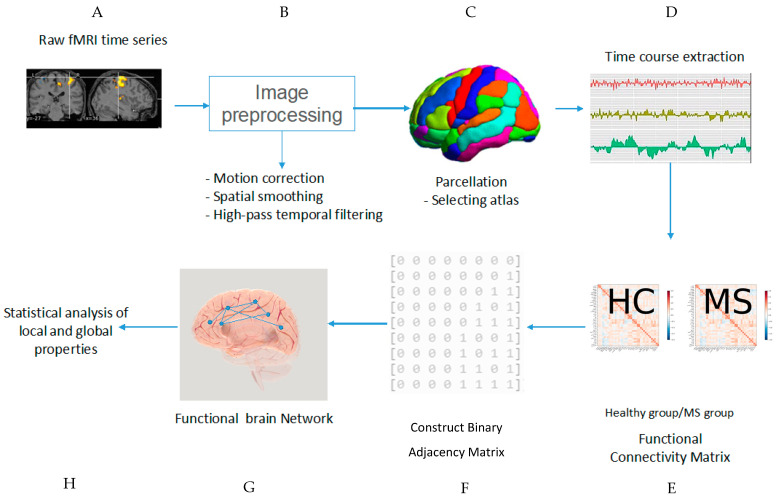
Pipeline of theoretical analysis using fMRI data. After collection of data with fMRI (**A**), brain image preprocessing (**B**) should be performed by applying motion correction, spatial smoothing, and high-pass temporal filtering. An atlas should be selected for dividing the entire brain into parcels (**C**). Time courses are extracted in the next level (**D**). The functional connectivity matrix and binary matrix are then constructed (**E**,**F**). After construction of the brain network, local and global features of the network can be calculated (**G**,**H**).

**Table 1 brainsci-13-00246-t001:** MS studies using graph theory.

Authors, Year	Number of Participants	Graph Theory Parameters	EDSS	Imaging/Cognitive Tests	Findings
[28]	330 (RRMS)	- Small-worldness - Global efficiency - Local efficiency	-	RS-fMRI	Decrease in regional efficiency for the insula, precentral gyrus, and prefrontal and temporal association cortices
[29]	16 (MS) 20 (HC)	- Modularity	Less than 2.5	PASAT	Increase in network modularity values for patients with MS
[30]	60 (MS) 59 (HC)	- Small-worldness - Centrality	-	RS-fMRI	Fatigue in patients with MS causing decreased in degrees in the bilateral thalamus; decreased degree in the bilateral caudate nucleus in non-fatigued patients
[31]	128 (MS) 50 (HC)	- Eigenvector centrality	Median 2.0	RS-fMRI	Increase in eigenvector centrality in the bilateral thalamus and posterior cingulate areas; decrease in eigenvector centrality in sensorimotor and ventral stream areas in patients with MS
[32]	246 (MS) 55 (HC)	- Global efficiency - Small-worldness	Less than 3.0	PASAT	Decrease in the nodal degree in the bilateral caudate nucleus and right cerebellum in patients with MS
[33]	38 (MS) 42 (HC)	- Nodal efficiency - Local efficiency - Connectivity index	0–7.0	RS-fMRI	Decrease in brain functional connectivity enhancement after advanced disease and during disease progression
[34]	18 (RRMS) 25 (HC)	- Clustering coefficient - Modularity	-	RS-fMRI	Increase in modularity in the brain network in patients with MS; increase in clustering coefficient and local efficiency
[35]	25 (MS) 35 (HC)	- Modularity	-	RS-fMRI	Increase in modularity in patients with MS
[36]	332 (MS) 96 (HC)	- Degree centrality	Median 3.0	RS-fMRI	Centrality increase in the default-node network in patients with MS
[37]	37 (PPMS) 21 (HC)	- Global efficiency - Local efficiency - Betweenness centrality - Characteristic path length	Less than 7.00	T25FW NHPT SDMT	With the definition of the rich club, rich club connectivity decreases among patients with MS
[38]	34 (MS) 34 (CIS) 36 (HC)	- Local efficiency - Global efficiency	0–6.5	PASAT	Decrease in brain network efficiency
[39]	46 (RRMS)	- Global efficiency - Assortativity - Characteristic path length - Modularity - Rich club coefficient - Transitivity	Median 2	RS-fMRI	Weaker connectivity strength, decrease in network density, reduction of global changes
[40]	41 (CIS) 32 (MS) 35 (HC)	- Modularity - Intra-module efficiency - Inter-module efficiency	0–6.5	PASAT2,3	No differences between CIS and HC; decrease in inter-module efficiency between sensory-motor network and frontoparietal network and between visual network and frontoparietal in MS
[41]	29 (MS) 24 (HC)	- Global efficiency - Local efficiency - Betweenness centrality - Characteristic path length - Clustering coefficient	3.2 ± 1.3	SDMT	Changes in serotonergic, noradrenergic, and cholinergic network functional connectivity in MS
[42]	37 (MS) 23 (HC)	- Clustering coefficient - Characteristic path length - Global efficiency	-	PASAT ANT SDMT MFIS NFI-MS BDI-II PSQI	Increase in modularity, increase in clustering and decrease in global efficiency in functional networks, and longer average path length
[43]	33 (RRMS) 29 (HC)	- Global efficiency - Characteristic path length - Clustering coefficient - Nodal strength	Median 2 0–4	SDMT PASAT TAP	Increase in functional connectivity due to maladaptive process
[44]	119 (MS) 42 (HC)	- Global efficiency - Degree centrality	2 (0–7.5)	PASAT	Alteration in the brain network in patients with MS according to disease duration
[45]	41 PWCIS 19 (HC)		Median EDSS=1 Start: 0–3 After 1 year: 0–5	RS-fMRI	No global efficiency difference between CIS and HC
[46]	25 (RRMS) 18 (HC)	- Global efficiency	Mean 4.3	CVLT-II, PASAT, SDMT, JLOT, COWAT	Decreased global efficiency in patients with MS with cognitive impairment
[47]	50 (MS) 26 (HC)	- Number of edges - Characteristic path length - Local efficiency - Global efficiency - Transitivity - Small-worldness	Less than 6.0	SDMT CVLT-II BVMT-R COWAT	No significant link between brain network alterations and cognition
[48]	32 (MS) 10 (HC)	- Centrality (degree and betweenness centrality) - Clustering coefficient - Characteristic path length	0–3.0	RS-fMRI	Significant increase in structural-functional coupling among patients with MS after 5 years
[49]	8 (MS) 12 (HC)	- Centrality - Modularity - Transitivity - Small-worldness	0–3.5	PASAT	Best sensitivity in MS analysis acquired from two global values of modularity and small-worldness index
[50]	32 (MS) 10 (HC)	- Strength - Centrality - Clustering coefficient	First episode suggestive of MS	SRT BVMT-R	Decrease in verbal memory function associated with hippocampal volume loss
[51]	10 (RRMS)	- Local efficiency - Global efficiency	-	RS-fMRI	Mixed observations in graph properties for walking exercise
[52]	48 (PPMS)	- Density - Strength - Local efficiency - Modularity	-	RS-fMRI	No significant difference for MS patients with and without optic neuritis in functional graph metrics.

MS: Multiple sclerosis, HC: healthy control, EDSS: Expanded Disability Status Scale, RS: resting state, RRMS: remitting/relapsing MS, PASAT: Paced Auditory Serial Addition Test, T25FW: timed 25-foot walk, NHPT: nine-hole peg test, SDMT: Symbol Digit Modalities Test, ANT: Attention Network Test, MFIS: Modified Fatigue Impact Scale, BDI-II: Beck Depression Inventory 2nd edition, PSQI: Pittsburgh Sleep Quality Index, COWAT: Controlled Oral Work Association Test, JLOT: Judgment of Line Orientation Test, CVLT: California Verbal Learning Test-II, BVMT-R: Brief Visuospatial Memory Test Revised, TAP: Test battery of Attentional Performance.

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
