# Peer review of "Graph-Based Analysis of Brain Connectivity in Multiple Sclerosis Using Functional MRI: A Systematic Review"

_brainsci, 2023, doi:10.3390/brainsci13020246_

Round 1

Reviewer 1 Report

The manuscript entitled “Graph-based Analysis of Brain Connectivity in Multiple Sclerosis Using Functional MRI: A Systematic Review” has been investigated in detail. The topic addressed in the manuscript is potentially interesting and the manuscript contains some practical meanings, however, there are some issues which should be addressed by the authors:

1)    In the first place, I would encourage the authors to extend the abstract more with the key results. As it is, the abstract is a little thin and does not quite convey the interesting results that follow in the main paper. The "Abstract" section can be made much more impressive by highlighting your contributions. The contribution of the study should be explained simply and clearly.

2)    The “Introduction” section needs a major revision in terms of providing more accurate and informative literature review and the pros and cons of the available approaches and how the proposed method is different comparatively. Also, the motivation and contribution should be stated more clearly.

3)    What makes the proposed method suitable for this unique task? What new development to the proposed method have the authors added (compared to the existing approaches)? These points should be clarified.

4)    “Discussion” section should be edited in a more highlighting, argumentative way. The author should analysis the reason why the tested results is achieved.

5)    The authors should clearly emphasize the contribution of the study. Please note that the up-to-date of references will contribute to the up-to-date of your manuscript. The study named "Artificial intelligence-based robust hybrid algorithm design and implementation for real-time detection of plant diseases in agricultural environments" - can be used to explain the proposed method in the study or to indicate the contribution in the “Introduction” section.

6)    Figure 2 should be improved.

7)    It will be helpful to the readers if some discussions about insight of the main results are added as Remarks.

Reviewer 2 Report

This systematic review did a literature search on the use of graph theory in fMRI studies on multiple sclerosis. While the literature search appeared comprehensive, the analysis and narrative parts can be further improved.

1. Can meta-analysis approaches be applied to the graph theory metrics across studies? Resting state vs. task-driven studies may need to be separated.

See related work on meta-analysis of fMRI studies.

https://www.ncbi.nlm.nih.gov/pmc/articles/PMC4478294/

2. The conclusion part has the first few sentences describing the organization of the paper. Those can be moved to the Introduction to guide the readers.

3. More background information about graph theory applications in fMRI studies can be helpful in the introduction.

4. Analysis and discussion of moderating factors can be helpful. How does age affect the connectivity measures? How do the resting-state functional brain networks relate to individual traits, cognitive ability and genetic factors?

https://www.ncbi.nlm.nih.gov/pmc/articles/PMC7196369/

5. What are the technical challenges considering the fact that the human brain is a complex network that can be studies at multiple spatial and time scales? How reliable are the network measure results?

6. When using abbreviated terms, spelling out the terms in the Table/figure  (example: Table 1) legends can be helpfu.

7. How is your systematic review different from other review articles on this topic? What is the novel contribution here? For example,

https://www.sciencedirect.com/science/article/abs/pii/S0306452217307613

Round 2

Reviewer 1 Report

My comments have been thoroughly addressed. 

Reviewer 2 Report

Minor errors can be fixed during proof stage. Please make sure that the sources are properly mentioned and corrected. For example, Line 482 shows Error! Reference source not found, the reference here should be Table 1.